# Icotinib, Almonertinib, and Olmutinib: A 2D Similarity/Docking-Based Study to Predict the Potential Binding Modes and Interactions into EGFR

**DOI:** 10.3390/molecules26216423

**Published:** 2021-10-24

**Authors:** Faisal A. Almalki, Ahmed M. Shawky, Ashraf N. Abdalla, Ahmed M. Gouda

**Affiliations:** 1Department of Pharmaceutical Chemistry, Faculty of Pharmacy, Umm Al-Qura University, Makkah 21955, Saudi Arabia; famalki@uqu.edu.sa; 2Science and Technology Unit (STU), Umm Al-Qura University, Makkah 21955, Saudi Arabia; amesmail@uqu.edu.sa; 3Central Laboratory for Micro-analysis, Minia University, Minia 61519, Egypt; 4Department of Pharmacology and Toxicology, Faculty of Pharmacy, Umm Al-Qura University, Makkah 21955, Saudi Arabia; anabdrabo@uqu.edu.sa; 5Department of Pharmacology and Toxicology, Medicinal And Aromatic Plants Research Institute, National Center for Research, Khartoum 2404, Sudan; 6Medicinal Chemistry Department, Faculty of Pharmacy, Beni-Suef University, Beni-Suef 62514, Egypt

**Keywords:** icotinib, almonertinib, olmutinib, similarity search, docking

## Abstract

In the current study, a 2D similarity/docking-based study was used to predict the potential binding modes of icotinib, almonertinib, and olmutinib into EGFR. The similarity search of icotinib, almonertinib, and olmutinib against a database of 154 EGFR ligands revealed the highest similarity scores with erlotinib (0.9333), osimertinib (0.9487), and WZ4003 (0.8421), respectively. In addition, the results of the docking study of the three drugs into EGFR revealed high binding free energies (Δ*G_b_* = −6.32 to −8.42 kcal/mol) compared to the co-crystallized ligands (Δ*G_b_* = −7.03 to −8.07 kcal/mol). Analysis of the top-scoring poses of the three drugs was done to identify their potential binding modes. The distances between Cys797 in EGFR and the Michael acceptor sites in almonertinib and olmutinib were determined. In conclusion, the results could provide insights into the potential binding characteristics of the three drugs into EGFR which could help in the design of new more potent analogs.

## 1. Introduction

Currently, many of the computational approaches are widely used in the discovery and development of new drug candidates [1,2]. Most of these approaches were supported by the advances in macromolecular X-ray crystallography. To date, more than 180,000 biological macromolecules are available on the protein data bank. These crystals provide crucial information about the druggable proteins and their co-crystallized ligands, which were used in the discovery of new drug candidates [3,4].

Among these computational techniques, the 2D similarity search is the fast and popular similarity search [5]. It includes different methods such as substructure and fingerprint similarity search. To quantify the degree of similarity between molecules, several metrics such as Tanimoto, Dice, and Cosine coefficient are being used [6]. Among these metrics, Tanimoto coefficient [7] was the most popular.

Molecular docking is one of the most commonly used computational techniques in medicinal chemistry and drug design [8]. Evaluation of the binding affinity, and interactions are among the valuable results that could be obtained from molecular docking [9]. In addition, the prediction of the binding modes of the new drug candidates could also be achieved using molecular docking studies [10].

Different computational techniques were also applied in the production of the rationally designed kinase inhibitors [11,12]. Targeting the oncogenic kinases was in focus in the last two decades supported by the high rates of cancer incidence and mortality [13]. Among these kinases, the epidermal growth factor receptor (EGFR) attracted considerable attention as a valid target in cancer chemotherapy [14]. Since the approval of the first EGFR inhibitor (gefitinib) by the FDA in 2003, huge efforts were conducted using computational chemistry to design new kinase inhibitors [15]. The great success in this area was confirmed by the development of many small molecules with potent inhibitory activities [15]. More than ten of these inhibitors were also approved as EGFR-targeting anticancer agents [16]. Among these inhibitors, icotinib (Figure 1) is a potent and selective EGFR inhibitor (IC_50_ = 5 nM) that was approved for the treatment of non-small cell lung cancer (NSCLC) [17]. In addition, almonertinib is a third-generation EGFR-TKI that was approved in 2020 in China for advanced EGFR T790M mutation-positive NSCLC [18,19]. Almonertinib showed higher inhibitory activities against several mutant EGFR than the wild-type EGFR.

Olmutinib (Figure 1) was also approved for the treatment of advanced EGFR T790M mutation-positive NSCLC in South Korea [20]. It acts as an irreversible inhibitor of EGFR, where it forms a covalent bond with the cysteine residue (Cys797) in the kinase. The kinase inhibition assay of olmutinib revealed inhibitory activity at an IC_50_ value of 0.01 μM against EGFR (790M/L858R) [21].

However, none of the three drugs (Figure 1) was co-crystallized with any of the target kinases. In addition, very little work was conducted to investigate their binding modes and interactions into their target kinases. Accordingly, investigation of the potential binding modes and interactions of the three drugs into EGFR could help in understanding their biological activities and could also support the design of more potent analogs in the future. Based on the previous reports, it was generally expected that in the same target, the ligands with high structural similarity exhibit similar binding interaction [22,23]. In the current study, a 2D similarity/docking-based study was used to predict the binding modes/interactions of each of the three drugs. The similarity search of the three drugs was performed against a database of 154 EGFR ligands obtained from the protein data bank (https://www.rcsb.org/ accessed on 28 July 2021). The ligand with the highest similarity (Tanimoto > 0.8) was identified and its binding mode into EGFR was used as a reference for the test compounds.

## 2. Results and Discussion

### 2.1. Similarity Search

Similarity search is a commonly used approach in drug research to identify a compound in the database with the highest similarity to the query molecules [24,25]. In the current study, the 2D similarity of the three drugs, icotinib, almonertinib, and olmutinib was evaluated against a database of 154 EGFR ligands obtained from the protein data bank (Appendix A). The similarity workbench in ChemMine tools (https://chemminetools.ucr.edu/) [26] was used to calculate the maximum common substructure (MCS) similarities between the compound pairs (MCS Tanimoto). The results in the form of similarity scores were used in the ranking of the compound database. The similarity search was performed for each of the three compounds, icotinib, almonertinib, and olmutinib, separately against EGFR-ligands. The search results were ranked in descending order to identify the ligand with the highest similarity. Discovery studio visualizer (v16.1.0.15350) [27] was used to generate the overlay figures of the tested drugs. Moreover, the molecular properties of these compounds were calculated using Molsoft L.L.C. (http://molsoft.com/mprop/ accessed on 7 August 2021) [28].

#### 2.1.1. Similarity Search for Icotinib

The results of the 2D similarity search of icotinib revealed the highest similarity score with erlotinib (MCS Tanimoto = 0.9333), Figure 2. Icotinib is a cyclized analog and is considered as a me-too drug of erlotinib [29]. An overlay view of the 3D representations of icotinib and erlotinib also showed superposition of the equivalent fragments in the two compounds, Figure 2.

The high similarity score between the two drugs was matched with their closely related molecular properties, Appendix A. The two compounds have comparable molecular volumes and closely related physicochemical properties. In addition, the number of hydrogen bond donors (HBDs) and acceptors (HBAs) are the same in the two molecules.

#### 2.1.2. Similarity Search for Almonertinib

The results of the similarity search of almonertinib revealed a similarity score of 0.9487 (MCS Tanimoto) with osimertinib, Figure 3. The high structural similarity score between almonertinib and osimertinib was matched with the structural analysis of the two compounds, whereby almonertinib could be obtained by replacement of the methyl group at N1 of the pyrrole ring in osimertinib by cyclopropyl moiety. Based on this small difference in the chemical structures of the two compounds, almonertinib can also be considered a “me too” drug of osimertinib.

The molecular properties of almonertinib and osimertinib were calculated using Molsoft molecular property prediction. The results are provided in the Appendix A. The results revealed a slight increase in molecular volume (~7%) that was observed in the replacement of the methyl group in osimertinib by cyclopropyl group in almonertinib. However, there was no change in the molecular polar surface area, number of HBAs, or HBDs. The two molecules are also active against EGFR T790M and are used in the treatment of EGFR T790M +ve NSCLC.

#### 2.1.3. Similarity Search for Olmutinib

The results of the similarity search of olmutinib also revealed a similarity score of 0.8421 (MCS Tanimoto) with WZ4003, an EGFR T790M inhibitor, Figure 4. The 5-chloropyrimidine in WZ4003 was replaced by thieno[3,2-*d*]pyrimidine in olmutinib. The molecular properties of olmutinib and WZ4003 were also calculated using Molsoft molecular properties prediction. The results are provided in the Appendix A. These results revealed nearly equal molecular volumes for the two compounds. The number of HBAs and HBDs are also identical in the two compounds. The two compounds are also effective against EGFR T790M.

### 2.2. Docking Study into EGFR

The docking studies of icotinib, almonertinib, and olmutinib were performed using AutoDock 4.1 [30]. Erlotinib, osimertinib, and WZ4003 were used as reference drugs in these studies based on the results of the 2D similarity search. Visualization of the binding modes of the three compounds was performed by discovery studio visualizer [27,31].

#### 2.2.1. Docking Study of Icotinib

The crystal structure of EGFR tyrosine kinase domain in complex with erlotinib (pdb: 1M17) [32] was selected for this study. Selection of this crystal was based on the high similarity score of icotinib and erlotinib (MCS Tanimoto = 0.9333). To validate the docking procedures, erlotinib was redocked into the active site of EGFR. The results revealed superposition of the redocked erlotinib with the co-crystallized ligand with RMSD of 1.45 Å, Figure 5.

The redocked erlotinib showed similar binding interaction with those of the co-crystallized erlotinib. Each of the two molecules exhibited one conventional hydrogen bond with the key amino acid Met769. In addition, similar hydrophobic interactions with the hydrophobic residues in EGFR were also observed, Figure 6.

To identify the equivalent binding interactions formed by the two molecules, LigPlot view was generated. The superposed LigPlot view revealed that most of the key binding interactions were preserved. Each of the two molecules exhibited one conventional hydrogen bond with Met769. In addition, the two molecules exhibited similar hydrophobic interactions with Ala719, Lys721, The766, Gln767, Leu768, Gly772, Leu694, Phe771, Leu820, and Asp831, Figure 6.

Icotinib was also docked into the crystal structure of EGFR (pdb: 1M17). The results revealed a binding affinity of −8.42 kcal/mol compared to −7.26 kcal/mol for erlotinib. The higher affinity of icotinib toward EGFR may be matched with its higher inhibitory activity against EGFR [17,33]. The binding modes/interactions of the top-scoring ten conformations of icotinib into EGFR were investigated to identify the pose which has similar orientation and binding interactions with the key amino acids like erlotinib. The results revealed that the seventh-ranked conformation superposed nicely over the co-crystallized erlotinib, where the quinazoline nuclei of the two compounds were located in the same plane and superposed each other, Figure 7.

The two ethynyl phenyl moieties in icotinib and erlotinib superposed each other into the back pocket of EGFR, whereby each moiety displayed one hydrophobic interaction with Lys721, Figure 7. The tetraoxacyclododecane ring in icotinib also adopted an orientation into the front pocket of EGFR like the two methoxyethoxy moieties in erlotinib. The similar orientation of the two molecules allowed the N2 in the quinazoline nucleus of icotinib to form one conventional H-bond with the key amino acid Met769. However, this hydrogen bond was longer (2.14 Å) compared to the corresponding hydrogen bond formed by erlotinib (1.64 Å), Figure 8.

Investigation of the binding interactions of icotinib into EGFR revealed similar binding interactions to those of co-crystallized erlotinib. The quinazoline nucleus in the two compounds also exhibited one carbon hydrogen bond with Gln767. Both icotinib and erlotinib also showed similar hydrophobic interactions with the hydrophobic residues in EGFR. To illustrate the equivalent interactions of the two compounds, a superposed LigPlot view of the two compounds was generated, Figure 9.

#### 2.2.2. Docking Study of Almonertinib

To investigate the potential binding mode of almonertinib, the crystal structure of the wild-type EGFR bound to the osimertinib (pdb: 4ZAU) inhibitor was used [34]. The selection of this crystal was based on the high similarity score (0.9487) between almonertinib and the co-crystallized ligand, osimertinib. The study was performed using AutoDock 4.2. Validation of the docking procedures was also performed whereby the co-crystallized ligand was redocked into the active site of EGFR. The results also revealed the superposition of the redocked osimertinib with the co-crystallized ligand with RMSD of 1.14 Å, Figure 10.

The binding interactions of the redocked osimertinib were also investigated in comparison with those of the co-crystallized ligand. Like the co-crystallized ligand, the redocked osimertinib exhibited two conventional hydrogen bonds with Met793 and Cys797. The two molecules also showed similar hydrophobic interactions with the same amino acids in EGFR, Figure 11.

To identify the equivalent binding interactions of the redocked and co-crystallized osimertinib, a superposed LigPlot view of the two molecules was generated, Figure 11. The figure showed that the two molecules have two equivalent hydrogen bonds with Met 793 and Cys797. In addition, similar hydrophobic interactions were formed by each of the two molecules with Leu792, Pro794, Leu718, Gly719, Val726, Ala743, and Gly796 amino acids in EGFR.

Almonertinib was docked into the active site of EGFR (pdb: 4ZAU) and the results obtained were compared with those of the co-crystallized osimertinib. The results revealed binding free energy of −6.32 kcal/mol compared to −7.03 kcal/mol for osimertinib. The top-scoring ten conformations of almonertinib were investigated to identify the best one which superposes with the co-crystallized osimertinib and exhibits similar binding interactions with the key amino acids in EGFR. Among these, the second-ranked pose showed superposition with the co-crystallized osimertinib and exhibited similar interactions with EGFR, Figure 12.

Investigation of the binding mode of the docked almonertinib revealed superposition of its pyrrole ring with the pyrrole ring in osimertinib. In addition, superposition of the pyrimidine rings in the two molecules was also observed, Figure 12.

Investigation of the binding interactions of almonertinib revealed three conventional hydrogen bonds with Met793 and Cys797. In addition, multiple hydrophobic interactions were also observed. To compare these interactions with those of the co-crystallized osimertinib, a LigPlot view of the two compounds was generated, Figure 13.

However, almonertinib and osimertinib are classified as third-generation EGFR inhibitor which also binds to the cysteine residue (Cys797) in EGFR kinase domain through a covalent bond [18,35]. Herein, the distances between the thiol group in Cys797 in EGFR and β-carbon of the Michael acceptor site in each of the two compounds were measured. The results revealed a distance of 4.765 Å for almonertinib compared to osimertinib (3.350 Å), Figure 14.

In conclusion, almonertinib exhibited lower binding free energy (Δ*G_b_* = −6.32 kcal/mol) compared to osimertinib (Δ*G_b_* = −7.03 kcal/mol). In addition, the β-carbon of the Michael acceptor site in almonertinib was located at a longer distance from the Cys797 in EGFR compared to osimertinib. However, almonertinib exhibited higher inhibitory activity against the wild-type EGFR [18,35]. This controversy could be attributed partially to the multi-target activity of the two compounds and to the different assay methods used in the biological evaluation of the kinase inhibitory activity of the two drugs.

#### 2.2.3. Docking Study of Olmutinib

In the docking study of olmutinib, the crystal structure of T790M- mutant EGFR (pdb: 5X2K) was used [36]. The selection of this crystal was based on the results of the similarity search which revealed the highest similarity (MCS Tanimoto = 0.8421) with WZ4003, the co-crystallized ligand in this crystal. Validation of the docking study was first done by redocking of WZ4003 into the EGFR T790M kinase domain. The results of the docking revealed the superposition of the redocked WZ4003 over the co-crystallized ligand with RMSD of 1.17 Å, Figure 15.

Investigation of the binding interactions of the co-crystallized WZ4003 revealed two conventional hydrogen bonds with the key amino acid Met793 with bond lengths of 1.62 and 2.11 Å. WZ4003 also exhibited multiple hydrophobic interactions Leu718, Val726, Ala743, Met790, Arg841, and Leu844, Figure 16. However, one unfavorable interaction of the acceptor-acceptor type was observed between the oxygen atom of the methoxy group in WZ4003 and the oxygen atom of the carbonyl group in Met793. The redocked WZ4003 also showed similar binding interactions with those of the co-crystallized WZ4003. One conventional hydrogen bonding interaction was observed between the NH group of the redocked WZ4003 and Met793 with bond length of 2.84 Å. In addition, the redocked WZ4003 also showed multiple hydrophobic interactions similar to those of the co-crystallized ligand.

To identify the equivalent binding interactions between the redocked and co-crystallized WZ4003, a superposed LigPlot view of the two molecules was generated, Figure 16. The figure illustrated that the two molecules have an equivalent hydrogen bond with Met 793. In addition, they showed equivalent hydrophobic interactions with Leu718, Ala743, Leu792, Pro794, and Gly796.

Olmutinib was also docked into the crystal structure of EGFR T790M (PDB: 5X2K) to identify the potential binding mode and interactions. The results of the docking study of olmutinib revealed a binding free energy of −8.40 kcal/mol compared to WZ4003 (−8.07 kcal/mol). The top ten scoring conformations of olmutinib were analyzed and their binding modes and interactions were compared with those of the co-crystallized WZ4003. The results also revealed that the sixth-ranked pose superposed with WZ4003 and formed two hydrogen bonds with Met793 and Cyc797, Figure 17.

The binding interactions of olmutinib were also visualized by LigPlot+, where two conventional hydrogen bonds with Met793 and Cys797 were present, Figure 18. In addition, a third hydrogen bond between N1 of the pyrimidine ring in olmutinib and Met793 was detected by LigPlot+. This hydrogen bond could account even partially for the higher binding free energy of olmutinib compared to WZ4003. The superposed LigPlot view of olmutinib and WZ4003 revealed two equivalent hydrogen bonds with Met793. The two molecules also exhibited equivalent hydrophobic interactions with Leu718, Ala743, Gln791, Leu792, Gly796, and Leu844.

Olmutinib was also reported to act as an irreversible EGFR mutant inhibitor [20]. The irreversible inhibition of EGFR depends on the presence of the Michael acceptor site which can bind covalently to the cysteine residue in EGFR. Accordingly, we investigated the binding mode of olmutinib to identify the location of the Michael acceptor site of olmutinib into the active site of EGFR T790M, Figure 19. The results revealed that the β-carbon of the acrylamide moiety in olmutinib is located at 4.087 Å away from the thiol group of Cys797 in EGFR compared to 4.120 Å for WZ4003.

## 3. Conclusions

During the last two decades, the discovery and development of EGFR inhibitors attracted a great attention of many researchers around the world. Among these inhibitors, icotinib, almonertinib, and olmutinib were approved for the treatment of cancers. However, they have not been yet co-crystallized with any of their target kinases. Accordingly, investigation of the potential binding affinities, modes, and interactions of the three drugs into EGFR could help in understanding the biological activities and support the design of more potent analogs. In the current study, a 2D similarity/docking-based study was performed to predict the potential binding modes of the three drugs. The 2D similarity of the three drugs were evaluated against a database of 154 EGFR ligands obtained from the protein data bank. Icotinib, almonertinib, and olmutinib showed similarity scores (MCS Tanimoto) of 0.9333, 0.9487, and 0.8421 with erlotinib, osimertinib, and WZ4003, respectively. The three drugs were docked into EGFR (pdb: 1M17, 4ZAU, and 5X2K), where the results revealed binding free energies (Δ*G_b_*) in the range of −6.32 to −8.42 kcal/mol compared with the co-crystallized ligands, erlotinib, Osimertinib, and WZ4003 (Δ*G_b_* = −7.03 to −8.07 kcal/mol). The top 10 scoring poses of each compound were analyzed to identify the potential binding poses which superposed with the reference ligand and exhibited equivalent interactions. Moreover, the distance between the thiol group of Cys797 in EGFR and the β-carbon in the Michael acceptor site in almonertinib and olmutinib were also determined (4.765 and 4.087 Å, respectively). In conclusion, the results of this study could provide insights into the binding characteristics of icotinib, almonertinib, and olmutinib into EGFR which could help in the understanding of their inhibitory activities as well as assisting in the design of new more potent analogs.

## 4. Experimental

### 4.1. Similarity Search

A database of 154 of the EGFR ligands were downloaded from the protein data bank, Appendix A. The isomeric smiles of these compounds were uploaded to ChemMine Tools (https://chemminetools.ucr.edu/ accessed on 26 August 2021) [26]. The similarity search was performed using Similarity workbench in ChemMine tools. The three drugs, icotinib, almonertinib, and olmutinib were uploaded (smiles) to the webpage. Each of the three drugs was compared with each compound in the database. The 2D similarity scores (MCS Tanimoto) were calculated and the results are provided in the Appendix A.

The molecular properties of icotinib, almonertinib, and olmutinib and the corresponding EGFR inhibitors, erlotinib, osimertinib, and WZ4003 were calculated using the molecular property prediction in Molsoft L.L.C. (http://molsoft.com/mprop/ accessed on 7 August 2021) [28]. The results are provided in Appendix A.

### 4.2. Molecular Docking

The docking study of icotinib, almonertinib, and olmutinib was performed in AutoDock 4.2 [30]. The docking study was performed into the crystal structure of EGFR (pdb: 1M17) [32], (pdb: 4ZAU) [34], and EGFR T790M (pdb: 5X2K) [36] of X-ray resolution 2.60, 2.80, and 3.20 Å, respectively. The crystal structures were downloaded from the protein data bank (https://www.rcsb.org/ accessed on 28 July 2021). Preparation of the ligand and protein files was done following the previous reports [37,38]. The docking procedures was performed following the previous reports [39,40]. A 3D grid box of 40 × 40 × 40 Å size (x, y, z) with the spacing of 0.375 Å centered at 22.01, 0.25, and 52.79 Å for docking into EGFR (pdb: 1M17). In addition, a 3D grid box of 45 × 45 × 45 Å size (x, y, z) with the spacing of 0.375 Å centered at −0.21, −50.29, and 17.98 Å for docking into EGFR (pdb: 4ZAU). Moreover, a 3D grid box of 45 × 45 × 45 Å size (x, y, z) with the spacing of 0.375 Å centered at −52.76, −1.11, and −20.6 Å for docking into EGFR (pdb: 5X2K). Validation of the docking study was performed by redocking of the co-crystallized ligand. Visualization of the binding modes and interactions of the tested compounds was done using Discovery Studio Visualizer [27] and LigPlot^+^ (v.2.1) [31].

## Figures and Tables

**Figure 1 molecules-26-06423-f001:**
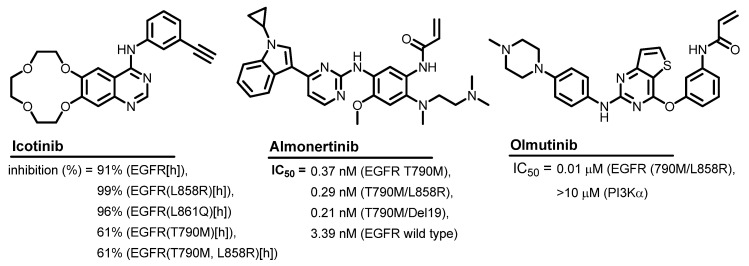
Icotinib, almonertinib, and olmutinib with their inhibition%/IC_50_ values against EGFR.

**Figure 2 molecules-26-06423-f002:**
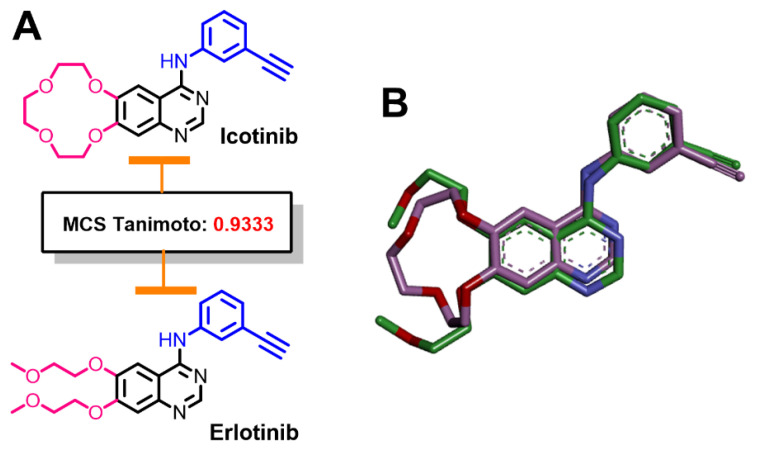
Structural similarity of icotinib and erlotinib: (**A**) 2D representations showing the equivalent fragments indicated by the same color; (**B**) overlay view of the two molecules (3D representations).

**Figure 3 molecules-26-06423-f003:**
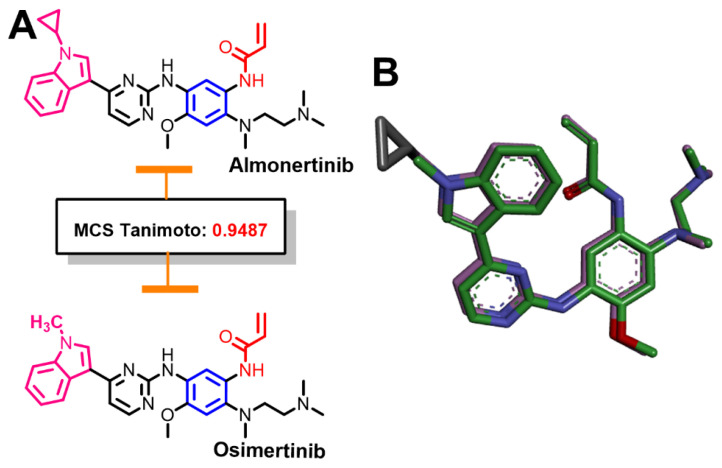
Structural similarity of almonertinib and osimertinib: (**A**) 2D representations showing the equivalent fragments indicated by the same color; (**B**) overlay view of the two molecules (3D representations).

**Figure 4 molecules-26-06423-f004:**
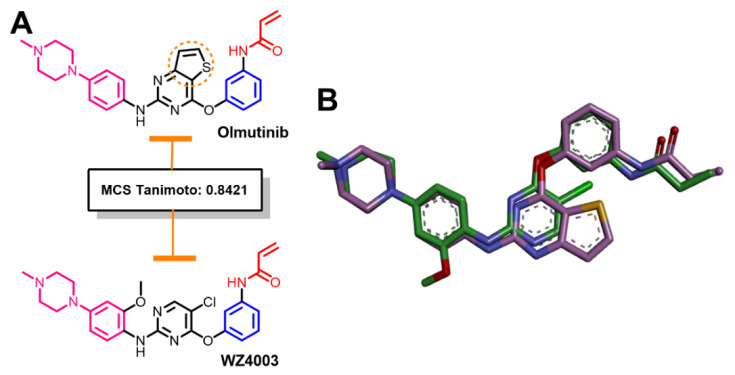
Structural similarity of olmutinib and WZ4003: (**A**) 2D representations showing the equivalent fragments indicated by the same color; (**B**) overlay view of the two molecules (3D representations).

**Figure 5 molecules-26-06423-f005:**
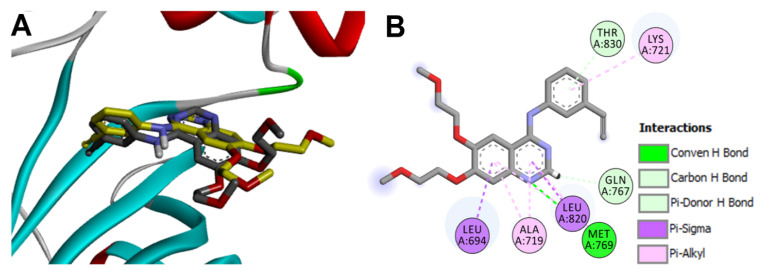
Binding mode/interactions of erlotinib (shown as sticks colored by element) into EGFR (pdb: 1M17): (**A**) 3D binding mode of the redocked erlotinib overlaid with the co-crystallized ligand (yellow sticks); (**B**) 2D binding mode of the co-crystallized erlotinib showing different types of binding interactions with amino acids in the active site of EGFR.

**Figure 6 molecules-26-06423-f006:**
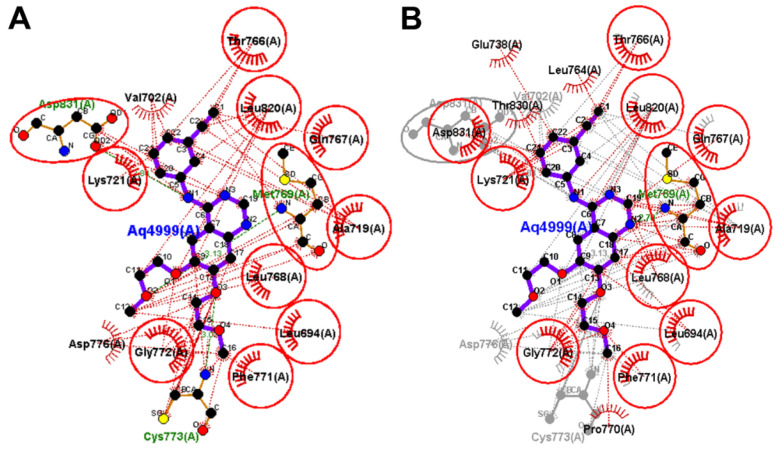
Binding interactions of erlotinib into EGFR (pdb: 1M17): (**A**) the redocked erlotinib; (**B**) superposed LigPlot view of the co-crystallized erlotinib (purple sticks), and the redocked erlotinib (gray sticks). The red circles/ ellipses show the equivalent binding interactions of the two molecules, hydrogen bond interactions are shown as olive green dotted lines and hydrophobic interactions are displayed in brick, red dotted lines.

**Figure 7 molecules-26-06423-f007:**
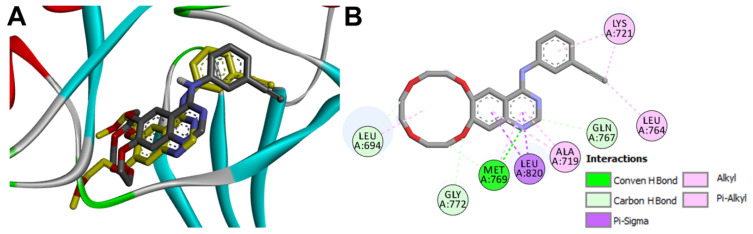
Binding mode/interactions of icotinib (sticks colored by element) into EGFR (pdb: 1M17): (**A**) 3D binding mode of icotinib overlaid with the co-crystallized ligand (yellow sticks); (**B**) 2D binding mode of icotinib showing different types of binding interactions with amino acids in the active site of EGFR.

**Figure 8 molecules-26-06423-f008:**
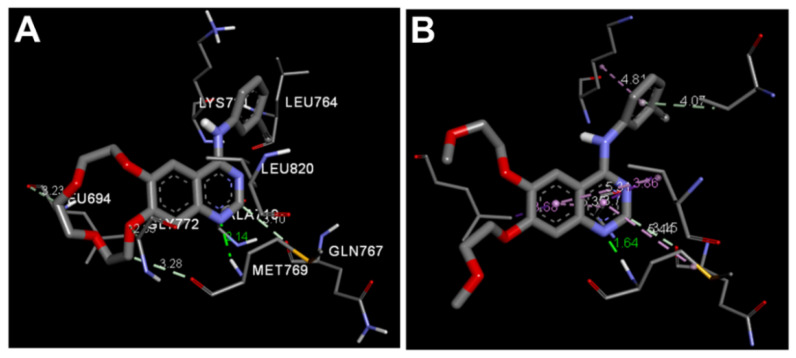
3D binding mode of erlotinib (**A**), and icotinib (**B**) showing the hydrogen bonds with their bond lengths (Å).

**Figure 9 molecules-26-06423-f009:**
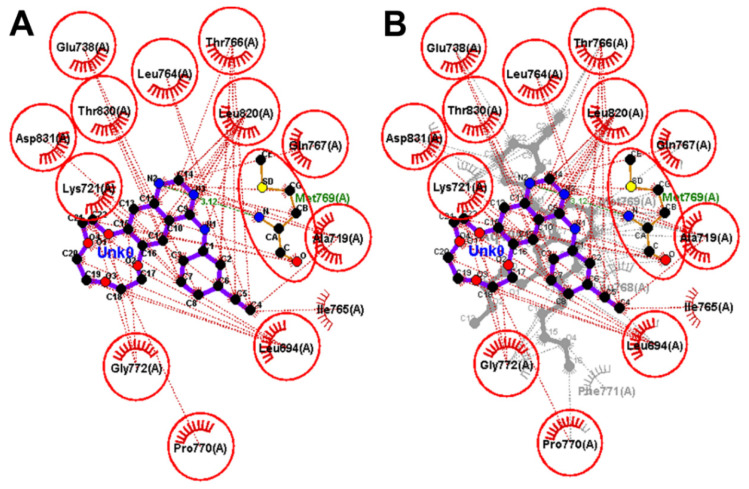
Binding interactions of the docked icotinib (purple sticks) into EGFR (pdb: 1M17): (**A**) icotinib; (**B**) superposed LigPlot view of icotinib (purple sticks) and the co-crystallized erlotinib (gray sticks), the red circles/ ellipses show the equivalent binding interactions, hydrogen bond interactions are displayed in olive green color dotted lines (Met769) and hydrophobic interactions are shown in brick red dotted lines.

**Figure 10 molecules-26-06423-f010:**
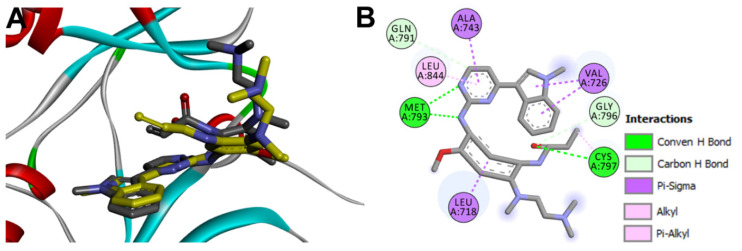
Binding mode/interactions of osimertinib into the active site of EGFR (pdb: 4ZAU): (**A**) 3D binding mode of the redocked osimertinib (shown as sticks colored by element) overlaid with the co-crystallized ligand (yellow sticks); (**B**) 2D binding mode of the co-crystallized osimertinib showing the binding interactions with amino acids in the active site of EGFR.

**Figure 11 molecules-26-06423-f011:**
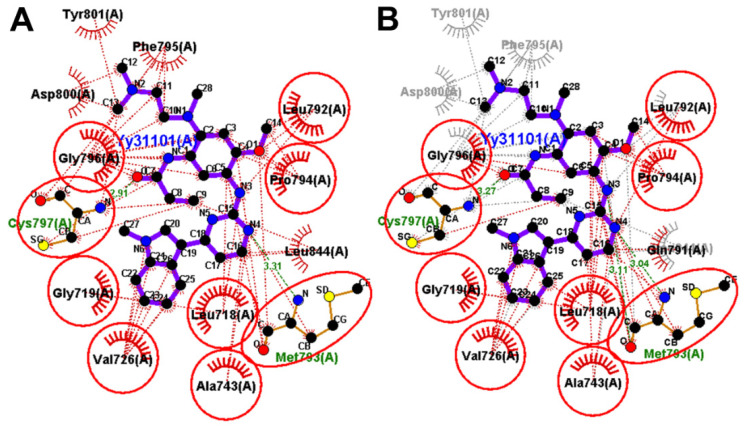
Binding interactions of osimertinib (shown as purple sticks) into EGFR (pdb: 4ZAU): (**A**) redocked osimertinib; (**B**) superposed LigPlot view of the co-crystallized osimertinib (purple sticks) overlaid with the redocked osimertinib (gray sticks), the red circles/ ellipses show the equivalent binding interactions of the two molecules, hydrogen bonds are shown as olive green dotted lines and hydrophobic interactions are shown in brick red dotted lines.

**Figure 12 molecules-26-06423-f012:**
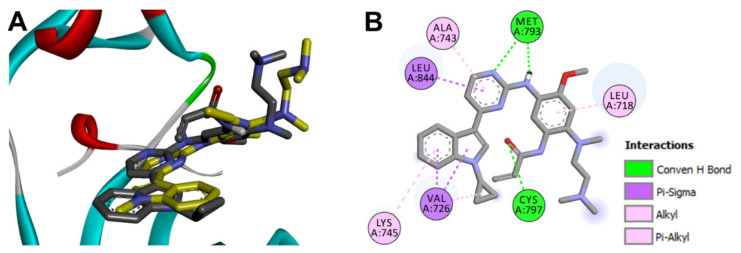
Binding mode/interactions of almonertinib (shown as ticks colored by element) into EGFR (pdb: 4ZAU): (**A**) 3D binding mode of the redocked almonertinib osimertinib overlaid with the co-crystallized ligand, osimertinib (shown as yellow sticks); (**B**) 2D binding mode of the redocked almonertinib showing different types of binding interactions with amino acids in the active site of EGFR.

**Figure 13 molecules-26-06423-f013:**
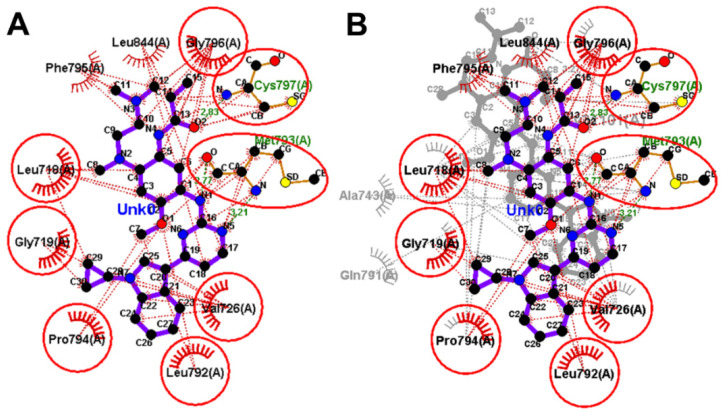
Binding interactions of almonertinib (shown as purple sticks) into EGFR (pdb: 4ZAU): (**A**) the docked almonertinib; (**B**) superposed LigPlot view of the docked almonertinib overlaid with the co-crystallized osimertinib (gray sticks), red circles/ ellipses show the equivalent interactions of the two molecules, hydrogen bond interactions are shown in olive green dotted lines (Met793, and Cys797) and hydrophobic interactions are shown in brick red dotted lines.

**Figure 14 molecules-26-06423-f014:**
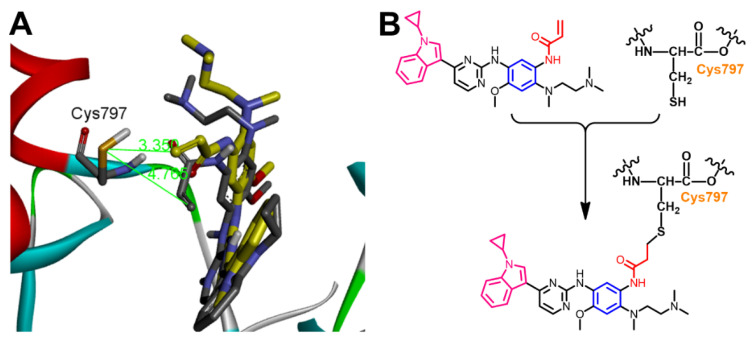
(**A**) 3D view of almonertinib (sown as sticks colored by element) overlaid with the co-crystallized osimertinib, showing the distance between the β-carbon of Michael acceptor sites and the thiol group of Cys797; (**B**) proposed mechanism of the covalent interaction between almonertinib and EGFR.

**Figure 15 molecules-26-06423-f015:**
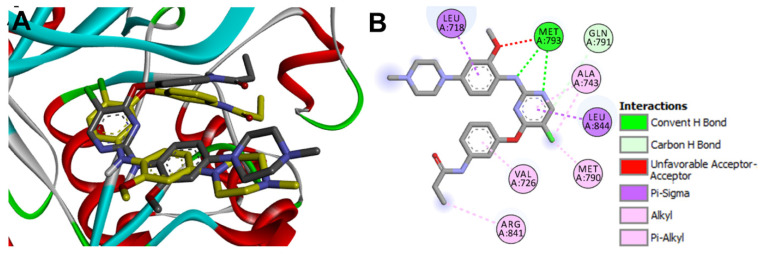
Binding mode/interactions of WZ4003 (shown as ticks colored by element) into EGFR T790M (pdb: 5X2K): (**A**) 3D binding mode of the redocked WZ4003 overlaid with the co-crystallized ligand, WZ4003 (yellow sticks); (**B**) 2D binding mode of the co-crystallized WZ4003 showing different types of the binding interactions with amino acids in the active site of EGFR.

**Figure 16 molecules-26-06423-f016:**
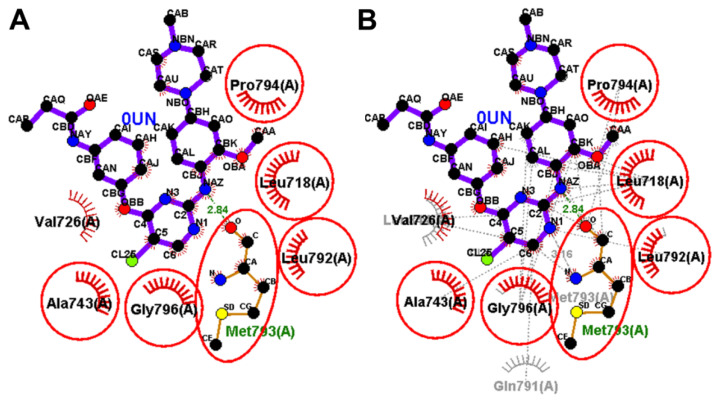
Binding interactions of the redocked WZ4003 (shown as purple sticks) into EGFR T790M (pdb: 5X2K): (**A**) redocked WZ4003; (**B**) superposed LigPlot view of the redocked WZ4003 (purple sticks) overlaid with the co-crystallized ligand (gray sticks), the red circles/ellipses show the equivalent binding interactions of the two molecules, hydrogen bond is shown in olive green dotted lines (Met793), and hydrophobic interactions are shown in brick red dotted lines.

**Figure 17 molecules-26-06423-f017:**
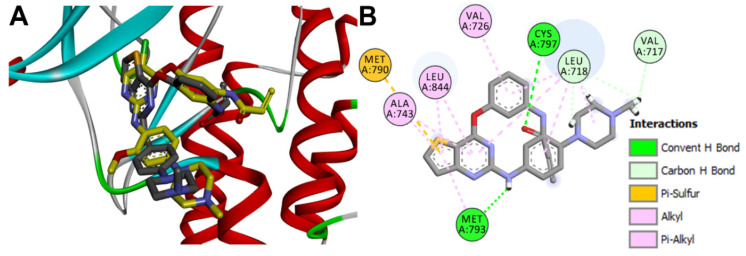
Binding mode/interactions of olmutinib (shown as sticks colored by element) into EGFR 696-1022 T790M (pdb: 5X2K): (**A**) 3D binding mode of olmutinib overlaid with the co-crystallized ligand, WZ4003 (yellow sticks); (**B**) 2D binding mode of olmutinib showing different types of binding interactions.

**Figure 18 molecules-26-06423-f018:**
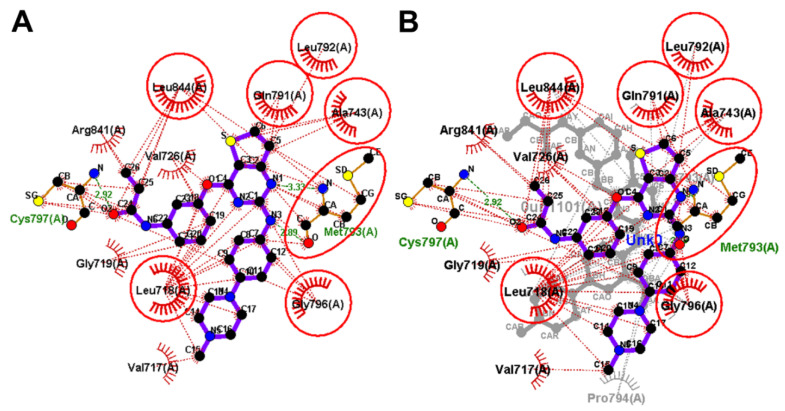
Binding interactions of olmutinib (shown as purple sticks) into EGFR T790M (pdb: 5X2K): (**A**) the docked olmutinib; (**B**) superposed LigPlot view of olmutinib and the co-crystallized WZ4003 (gray sticks), the red circles/ellipses show the equivalent binding interactions, hydrogen bonds are shown in olive green dotted lines, and the hydrophobic interactions are shown in brick red dotted lines.

**Figure 19 molecules-26-06423-f019:**
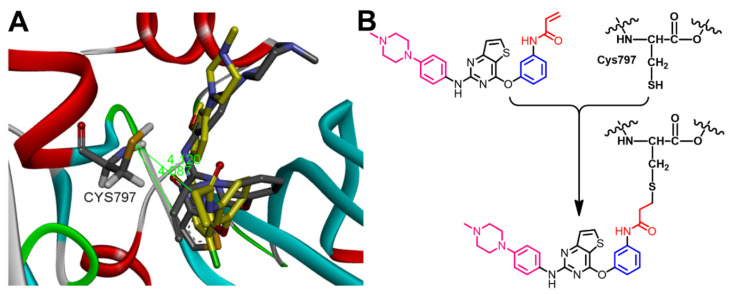
(**A**) 3D view of olmutinib (shown as sticks colored by element) overlaid with the co-crystallized WZ4003, showing the distance between β-carbon of Michael acceptor sites and the thiol group of Cys797; (**B**) proposed mechanism of the covalent interaction between olmutinib and EGFR.

## Data Availability

Data regarding this article will be provided upon request.

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
