# Peer review of "Icotinib, Almonertinib, and Olmutinib: A 2D Similarity/Docking-Based Study to Predict the Potential Binding Modes and Interactions into EGFR"

_molecules, 2021, doi:10.3390/molecules26216423_

Round 1
Reviewer 1 Report
The manuscript of Faisal A. Almalki et al. “Icotinib, almonertinib, and olmutinib: A 2D similarity/docking-based study to predict the potential binding modes and interactions into EGFRs” is devoted to the 2D similarity/docking-based study to predict the potential binding modes of icotinib, almonertinib, and olmutinib into EGFR. The analysis of the top-scoring poses of the three drugs was done to identify their potential binding modes with the EGFR.
In general, the manuscript is well organized and documented, and written in acceptable form.
Thus, I would like to recommend few corrections:
Page 1
Lines 41-42: Please rewrite this sentence “Molecular docking, another computational technique that was also extensively used in medicinal chemistry and drug design” as it seems to be without a logical ending.
Page 2
Figure heading states “… IC50 values against EGFR”, although there is an inhibition of EGRF by icotinib in % and no IC50 values are presented. Either the Figure heading or the data presented should be changed.
Page 4
Line 124: “…being observed” or “…that was observed…”
Line 132: “…in olmutinib..”
Page 9
Lines 251-252: it is not clear what did the authors mean by “..is located in the same plan of the and the 1-methyl-1H-indole moiety of osimertinib…”
Page 10
Lines 278: “… be attributed to..”
Page 13
Lines 351: “… attention of many..”
My decision is a minor revision.
Author Response
The manuscript of Faisal A. Almalki et al. “Icotinib, almonertinib, and olmutinib: A 2D similarity/docking-based study to predict the potential binding modes and interactions into EGFRs” is devoted to the 2D similarity/docking-based study to predict the potential binding modes of icotinib, almonertinib, and olmutinib into EGFR. The analysis of the top-scoring poses of the three drugs was done to identify their potential binding modes with the EGFR.
Comment: In general, the manuscript is well organized and documented, and written in acceptable form.
We highly appreciate the valuable comments of reviewer 1 and his valuable corrections that have been emerged after his careful and precise revision which would help in improving the quality of the manuscript. In addition, we indicated the revisions/corrections by a yellow highlighter in the revised manuscript. Below, are our responses to the comments, point-by point.
Thus, I would like to recommend few corrections:
Comment: Page 1: Lines 41-42: Please rewrite this sentence “Molecular docking, another computational technique that was also extensively used in medicinal chemistry and drug design” as it seems to be without a logical ending.
Response: We would like to thank the reviewer for this comment. Accordingly, the syntax of the indicated sentence was changed in the revised manuscript.
Comment: Page 2: Figure heading states “… IC50 values against EGFR”, although there is an inhibition of EGRF by icotinib in % and no IC50 values are presented. Either the Figure heading, or the data presented should be changed.
Response: Again, we would like to thank reviewer 1 for this observation. The legend of Figure 1 was changed in the revised manuscript.
Comment: Page 4: Line 124: “…being observed” or “…that was observed…”
Response: It was corrected in the revised manuscript.
Comment: Line 132: “…in olmutinib..”
Response: It was corrected in the revised manuscript.
Comment: Page 9: Lines 251-252: it is not clear what did the authors mean by “..is located in the same plan of the and the 1-methyl-1H-indole moiety of osimertinib…”
Response: The syntax of this sentence was paraphrased in the revised manuscript.
Comment: Page 10: Lines 278: “… be attributed to..”
Response: it was corrected in the revised manuscript.
Comment: Page 13, Lines 351: “… attention of many..”
Response: it was corrected in the revised manuscript.
Reviewer 2 Report
Almalki et al. have investigated the recognition mode of three relevant drugs targeting EGFR through a docking approach based on 2D similarity searches. The manuscript is well organized, a bit repetitive for each case study, but presented with enough clarity. The authors refer to supplementary material, however, it was not available for this reviewer, which could not evaluate the content there. Nevertheless, this reviewer considers this manuscript suitable for publication after minor revision.
English requires some editing. In several places, the verbs are missing or some words should be removed to make understandable the sentence. For instance, authors should revise the following sentences:
Page 1, line 41.
Page 2, lines 56 and 57.
Page 9, line 251.
Page 12, lines 326-327.
In the introduction, the authors should clarify which Cys of the kinase is involved in the recognition of the covalent inhibitors.
The authors should indicate the resolution of the crystallographic structures that they have used to perform the docking calculations.
Page 10, line 274: binding free energies values are missing for comparison reasons.
Author Response
Almalki et al. have investigated the recognition mode of three relevant drugs targeting EGFR through a docking approach based on 2D similarity searches. The manuscript is well organized, a bit repetitive for each case study, but presented with enough clarity. The authors refer to supplementary material, however, it was not available for this reviewer, which could not evaluate the content there. Nevertheless, this reviewer considers this manuscript suitable for publication after minor revision.
We highly appreciate the valuable comments of reviewer 2 on our manuscript, that have been emerged after his careful and precise revision which would help in improving the quality of the manuscript. The supplementary data of the 2D similarity study was uploaded with the revised manuscript. Below, are our responses to the other comments, point-by point.
Comment: English requires some editing. In several places, the verbs are missing, or some words should be removed to make understandable the sentence. For instance, authors should revise the following sentences:
Response: The manuscript was revised for the typo/grammar mistakes. All the modifications done were highlighted in yellow
Comment: Page 1, line 41.
Response: Response: the sentence in line 41 was revised and corrected per the reviewer’s suggestion.
Comment: Page 2, lines 56 and 57.
Response: the sentence in lines 56 and 57 was revised and corrected in the revised manuscript.
Comment: Page 9, line 251.
Response: the sentence in line 251 was revised and corrected in the revised manuscript.
Comment: Page 12, lines 326-327.
Response: the sentence in lines 326-327 was revised and corrected in the revised manuscript.
Comment: In the introduction, the authors should clarify which Cys of the kinase is involved in the recognition of the covalent inhibitors.
Response: The cysteine residue (Cys797) involved in the covalent interaction of olmutinib/osimertinib with EGFR was indicated in the introduction part in the revised manuscript.
Comment: The authors should indicate the resolution of the crystallographic structures that they have used to perform the docking calculations.
Response: The resolution of each of the three crystals used in the current study was indicated in the revised manuscript.
Comment: Page 10, line 274: binding free energies values are missing for comparison reasons.
Response: The ΔGb values of almonertinib and osimertinib were added in the revised manuscript.